# Tumor-Suppressive and Oncogenic Roles of microRNA-149-5p in Human Cancers

**DOI:** 10.3390/ijms231810823

**Published:** 2022-09-16

**Authors:** Yang Shen, Nan Zhao, Nan Zhao, Xinyao Hu, Xiaoqin He, Yangtao Xu, Jiayu Chen, Wenliang Chen, Xin Liu, Zhuolin Zhou, Dedong Cao, Ximing Xu

**Affiliations:** 1Department of Oncology, Renmin Hospital of Wuhan University, Wuhan 430060, China; 2Department of Endocrinology, Renmin Hospital of Wuhan University, Wuhan 430060, China

**Keywords:** microRNAs, miRNA-149-5p, tumor-suppressive factor, oncogenic factor, ionizing radiation, tumor immunity

## Abstract

Malignant tumors are always a critical threat to human health, with complex pathogenesis, numerous causative factors, and poor prognosis. The features of cancers, such as gene mutations, epigenetic alterations, and the activation and inhibition of signaling pathways in the organism, play important roles in tumorigenesis and prognosis. MicroRNA (miRNA) enables the control of various molecular mechanisms and plays a variety of roles in human cancers, such as radiation sensitivity and tumor immunity, through the regulation of target genes. MiR-149-5p participates in the process and is closely related to lipogenesis, the migration of vascular endothelial cells, and the expression of stem-cell-related proteins. In recent years, its role in cancer has dramatically increased. In this review, we summarize the regular physiological roles of miRNAs, specifically miR-149-5p, in the organism and discuss the tumor-suppressive or oncogenic roles of miR-149-5p in different human cancers with respect to signaling pathways involved in regulation. Possible clinical applications of miR-149-5p in future targeted therapies and prognosis improvement in oncology are suggested.

## 1. Introduction

MiRNAs are a class of non-coding RNAs widely found in living organisms, mainly located in exonic or intronic regions of non-coding RNAs and to a lesser extent in gene sequences encoding proteins. Some mature miRNAs themselves also contain relatively independent promoter regions [1,2]. MiRNAs can regulate signaling pathways in vivo through miRNA editing, interaction with mRNAs, interaction with endogenous competing RNAs (ceRNAs), or binding with RNA-binding proteins [3]. The role of miRNAs in cancer has received much attention in the last decade. The role of miRNAs in the regulation of apoptosis can be achieved directly through changes in the number of miRNA motifs, mutations in miRNA genes, and epigenetic changes (e.g., methylation and acetylation) [4,5,6]. It can also regulate the tumorigenesis, invasion, metastasis, and angiogenesis of cancers via transcriptional regulation, post-transcriptional modifications (i.e., recoding, chemical modifications, etc.) [7], and the dysregulation of miRNA processing. In addition, miRNAs are expressed at different levels in healthy individuals versus tumor patients [8]. By detecting differences in miRNA expression levels in body fluids, miRNAs can be used as biological tumor markers to improve the detection rate of early tumors and enrich the assessment of patient prognosis [9].

Furthermore, miRNAs can regulate the innate immunity to the tumor by regulating macrophage polarization, NK-cell proliferation, antigen presentation, and the immunogenicity of dendritic cells. MiRNAs can also specifically kill mutant or tumor cells through TNF-TNFR and Fas-FasL pathways. Studies have shown that miRNAs can be combined with drug carriers such as exosomes or nanoparticles to silence tumor-cell genes or regulate the tumor immune microenvironment (TIME) to promote the apoptosis of tumor cells [10,11]. In addition, miRNAs can also be used to regulate the expression of relevant genes in tumor cells through gene disruption, enhancing the sensitivity of tumor cells to anticancer drugs [12]. When faced with the “double-edged sword” of ionizing radiation (IR), some miRNAs participate in the carcinogenesis induced by ionizing radiation. In contrast, some miRNAs respond to the body’s damage and enhance tumor tissue’s sensitivity to ionizing radiation by participating in DNA damage repair and cell-cycle regulation and by affecting the expression of apoptosis-related genes to promote autophagy [13,14]. In conclusion, miRNAs are widely regarded as potential biomarkers for many diseases, especially cancer, and have attracted extensive attention.

MiRNA-149-5p, as a member of the miRNA family, has a wide range of physiological effects, such as promoting the generation of bovine adipocytes, regulating the function of vascular smooth muscle, and promoting the positive regulation of goat hair-follicle stem cells. The expression of miR-149-5p is also significantly changed in patients with various cancers, which indicates that it might play a dual role of tumor-suppressive and oncogenic factor. MiR-149-5p acts as a tumor-suppressive factor in hepatocellular carcinoma (HCC), gastric cancer (GC), colorectal cancer (CRC), and breast cancer (BC), while in lung adenocarcinoma (LUAD), acute lymphoblastic leukemia (ALL), and acute myelogenous leukemia (AML), it acts as a cancer-associated miRNA that promotes tumorigenesis and invasion. MiR-149-5p is regulated by long non-coding RNAs (lncRNAs) and circular RNAs (circRNAs) during cancer development, such as LINC00460 in CRC, circNRIP1 in GC, and hsa_circ_0075341 in cervical cancer. However, the role and mechanisms of miR-149-5p in cancer initiation and progression are not entirely understood [15]. Therefore, it is particularly important to pay attention to the normal physiological role of miR-149-5p in living organisms, construct the corresponding DNA–RNA–protein expression network, and analyze the role of miRNA-149-5p in different human cancers. This review summarizes miR-149-5p expression, function, target genes, upstream regulators in human cancers, and its potential application in various systemic cancers.

## 2. Physiological Roles of miRNA-149-5p

### 2.1. miRNA-149-5p and Adipogenesis

Khan and his team found that miR-149-5p played a role in bovine adipogenesis and regulated the expression levels of bovine adipocyte genes CCND2, KLF6, ACSL1, Cdk2, SCD, SIK2, and ZEB1 and their KEGG pathway activity [16].

Cancer-associated adipocytes (CAAs) are a group of adipocytes that exert pro-tumorigenic effects by influencing the secretion of adipokines and pro-inflammatory cytokines and are essential components of the tumor microenvironment [17]. A genome-wide analysis showed that miRNAs played an important role in the association between obesity and cancer in humans. Among them, miR-3184-5p and miR-181c-3p were key regulators of adipocyte-associated BC and achieved the response of BC cells to CAAs through targets FOXP4 and PPARα. Moreover, researchers found that there were 20 up-regulated miRNAs in BC responding to CAAs. Among the 20 up-regulated miRNAs, in addition to miR-3184-5p, the expression of miR-149-5p was also significantly up-regulated [18].

This study suggested that miR-149-5p in the human body may also lead to the occurrence of diseases by affecting the fat metabolism in the human body or promote the occurrence of tumors by regulating the function of CAAs. However, there is a lack of studies on miR-149-5p in human tissues, and the mechanism of miR-149-5p regulating fat metabolism is unclear and needs to be paid attention to. Once the relationship between miR-149-5p and human fat metabolism is proved, miR-149-5p can be used as one of the known targets of human fat metabolism to enrich the understanding of lipid metabolic diseases and tumor microenvironment (TME).

### 2.2. miR-149-5p and Vascular Smooth Muscle Cells

Vascular smooth muscle cells (VSMCs) play a crucial role in maintaining the normal structure of blood vessels. They can migrate to the intima to replenish neointimal damage when a pathological injury occurs [19]. Zhang et al. found that miR-149-5p was down-regulated in PDGF-BB-induced VSMCs. The overexpression of miR-149-5p inhibited the proliferation, invasion, and migration of VSMCs, and the knockdown of miR-149-5p gave the opposite result. HDAC4, a potential target of miR-149-5p, could rescue the inhibitory effect of miR-149-5p on VSMC inhibition [20]. Peng et al. found that the CircDHCR24/miR-149-5p/MMP9 axis was involved the PDGF-BB-induced proliferation, migration, and phenotypic transformation of VSMCs [21]. Moreover, Wang et al. found that circ_CHFR could promote the proliferation, migration, and invasion of human VSMCs by sponging miR-149-5p and regulating NRP2 expression [22].

### 2.3. miR-149-5p and Biological Stem Cells

Wang et al. found that miR-149-5p was highly expressed in the skin tissues of Yangtze Delta white goats with high-quality bristles, whereas the CMTM3 gene was lowly expressed. MiR-149-5p promoted the proliferation and inhibited the apoptosis of hair-follicle stem cells by directly targeting CMTM3, thus achieving the positive regulation of hair-follicle stem cells in goats [23].

As for studies of biological stem cells in humans, there is no relevant study of miR-149-5p. However, in vitro experiments showed that CMTM3 could inhibit the osteogenic differentiation of mesenchymal stem cells [24]. The linkage of miR-149-5p with CMTM3 and its downstream pathways should be further searched to explore the potential role of miR-149-5p in human stem cells.

## 3. miR-149-5p in Different Human Cancers

### 3.1. Cancers of the Endocrine System

#### Thyroid Cancer

Thyroid cancer (TC) is a common form of endocrine cancer. There are three main histological types of TC: differentiated (papillary and follicular TC), undifferentiated (hypodifferentiated and mesenchymal TC), and medullary TC of parafollicular origin.

The prognosis of papillary thyroid cancer (PTC) is better, but the molecular mechanism of PTC remains unclear [25]. Rs2292832 is a genetic polymorphism located in the miR-149 precursor. In patients with PTC, CC genotype rs2292832 differs from the TC and TT genotypes and is significantly associated with an increased risk of tumorigenesis and invasion in PTC. One study found that miR-149-5p was expressed at lower levels in PTC patients with the CC genotype than the other two genotypes, suggesting that miR-149-5p may play a tumor-suppressive role in PTC. At the same time, rs2292832 may be involved in the susceptibility and local progression of PTC in Chinese patients by changing the expression levels of miR-149-5p and its target genes [26]. Ouyang et al. found that TR4 expression was higher in PTC patients with distant metastases than in patients without distant metastases. It could bind to the upstream promoter of circFLNA to regulate its expression at the transcriptional level. Meanwhile, circFLNA could target miR-149-5p to upregulate MMP9 expression and promote PTC-cell invasion and migration and was validated in a mouse xenograft transplantation tumor model [27].

Sun et al. found that chemokine CCL18 expression was significantly increased in the tissues of TC patients and was closely associated with the clinical stage of TC and the occurrence of lymph-node metastasis. They also demonstrated that CCL18 was a potential downstream target gene of miR-149-5p in TC, but the exact mechanism by which miR-149-5p regulated CCL18 expression and thus affected TC was unclear [28]. Peng and his team found that the circ-CSNK1G1/miR-149-5p/MAPK1 axis played an important regulatory role in TC, with circ-CSNK1G1 competing with MAPK1 for miR-149-5p binding sites and mitigating the inhibitory effect of miR-149-5p on MAPK1, thereby promoting tumorigenesis [29].

Medullary thyroid cancer (MTC) is less common but has a poor prognosis. Ye et al. demonstrated using cellular assays that miR-149-5p inhibited the proliferation and invasion of MTC cells by targeting GIT1 [30].

Although the exact mechanism of miR-149-5p regulation in TC is unclear, the available findings confirm that miR-149-5p acts as an oncogenic repressor in TC. Upstream circRNAs and downstream target genes often regulate this function. How to explore the mechanism of miR-149-5p regulating TC is one of the most important research issues at present. MiR-149-5p has great potential to be a staging and prognostic indicator of TC (Figure 1).

### 3.2. Hematologic Malignancies

Leukemia is one of the common neoplasms of the hematological system and is classified into AML, ALL, chronic myelogenous leukemia (CML), chronic lymphocytic leukemia (CLL), and other types of leukemia.

Tian et al. found that miR-149-5p expression was significantly upregulated in blood and bone marrow samples from AML patients and AML cell line THP-1. The overexpression of miR-149-5p decreased FASLG expression and activated FADD and cystathionase, thereby promoting the apoptosis of AML cells [31]. Zhu et al. found that hsa_circ_0120872 expression was significantly downregulated in pediatric ALL patients. They named hsa_circ_0120872 circADD2 and verified that circADD2 bound to protein AGO2 and miR-149-5p to form a ternary complex that downregulated the expression of miR-149-5p target gene AKT2, thus inhibiting the proliferation and inducing the apoptosis of ALL cells [32,33].

Although the role of miR-149-5p in other types of hematologic malignancy remains unclear, the tumor suppressor role of miR-149-5p in AML and pediatric ALL has been established. Considering that miR-149-5p is upregulated in the tissues and blood of AML patients, miR-149-5p in combination with existing diagnostic markers could possibly improve the early detection rate of AML and provide patients with more timely treatment. Moreover, miR-149-5p is a promising therapeutic target in hematological malignancies, and more studies are needed to enrich the understanding of miR-149-5p and its corresponding regulatory pathways (Figure 1).

### 3.3. Cancers of the Central Nervous System (CNS)

#### 3.3.1. Glioma

CNS cancers often significantly impact patients’ lives, with 25,050 new confirmed cases and 18,280 deaths predicted in the United States alone in 2022 [33].

Gliomas are a relatively common group of tumors. Gliomas are a relatively common group of neurological tumors. Xu et al. found decreased miR-149-5p expression in glioma cell lines, tumor tissue, and leukocytes from glioma patients. However, in another glioma study, glioma patients with the RS2292832 CC/CT genotype had increased levels of miR-149-5p expression and higher overall and disease-free survival rates. Moreover, glioma patients with the CC/CT genotype could achieve better therapeutic outcomes via the increase in the expression level of miR-149-5p, thus promoting the cytotoxic effect of temozolomide on glioma [34].

This indicates that miR-149-5p may also play different physiological roles by regulating different molecular pathways in the same type of tumor cells. More studies are needed to explore its molecular pathways and increase the understanding of miR-149-5p in the occurrence, treatment, and prognosis of glioma and even CNS cancers.

#### 3.3.2. Pituitary Cancers

Pituitary tumors are also a group of primary intracranial tumors with high incidence, among which invasive pituitary adenomas (IPAs) still lack effective treatment [35].

Zhao et al. found that miR-149-5p expression was significantly downregulated in IPA patients. Exosomes carrying miR-149-5p inhibited cell proliferation, invasion, metastasis, and angiogenesis in IPA tumor cells, confirming that miR-149-5p is an oncogenic repressor factor in IPAs [36].

At present, the understanding of IPAs is not comprehensive, and the treatment is single. MiR-149-5p may be used as a therapeutic target of IPAs to effectively inhibit tumor cells through drugs or exosome technology.

### 3.4. Bone Cancers

Chondrosarcoma is a common malignant tumor of soft tissue and bone that is highly susceptible to metastasis to distant organs such as the lungs, severely affecting patient prognosis [37].

Tzeng et al. found that in human chondrosarcoma cells, miR-149-5p controlled the expression of lysyl oxidase (LOX) through the PI3K/Akt/mTOR pathway, thereby inhibiting the development and metastasis of chondrosarcoma. However, the oncogenic repressor effect of miR-149-5p was inhibited by nerve growth factor (NGF) [38].

MiR-149-5p may be one of the targets to inhibit the occurrence and metastasis of chondrosarcoma, and the specific mechanism of NGF’s inhibitory effect on the anti-cancer effect of miR-149-5p is still unclear, which is worthy of further investigation and study.

### 3.5. Cancers of the Digestive System

#### 3.5.1. Hepatocellular Carcinoma

HCC is the fourth most common cancer worldwide, with limited treatment options and poor prognosis [39].

Many studies focused on the regulation of miR-149-5p via its upstream lncRNAs in HCC. Dong et al. found that lncRNA SNHG8 sponged miR-149-5p to promote HCC-cell proliferation, invasion, and lung metastasis [40]. Ji et al. found that LINC00461 promoted the proliferation, migration, and invasion of HCC cells by sponging miR-149-5p and up-regulating the expression of LRIG2, the downstream target gene of miR-149-5p [41]. Zhou et al. found that miR-149-5p expression was lower in HCC tissues than in normal tissues and negatively regulated MAP2K1. LncRNA PART1 sponged miR-149-5p, weakening its inhibition of MAP2K1 and promoting HCC-cell proliferation, migration, and invasion [42].

All of the above studies suggested that miR-149-5p played a tumor-suppressive role in HCC, while lncRNA promoted HCC progression by suppressing miR-149-5p expression. Therefore, targeting these lncRNAs and elevating miR-149-5p levels could be a potential strategy to block HCC progression. Fu and his colleagues confirmed this; they found that in HCC, aloin could target the circ_0011385/miR-149-5p/WT1 axis to exert anti-tumor effects [43]. Although aloin is a targeting substance concerning circRNAs, this study opened up additional possibilities for the similar targeting of lncRNAs that regulate the expression of miR-149-5p in HCC.

In addition to lncRNAs, Liu et al. found that M2-type macrophages could inhibit miR-149-5p expression to promote HCC progression [44]. M2-type macrophages are key immunosuppressive cells in the TIME, and this finding further confirmed the tumor-suppressive roles of miR-149-5p in HCC. It is widely recognized that the TIME is crucial in tumor growth, development, and metastasis. However, there is a gap in research on the roles of miR-149-5p in the TIME. More studies should focus on this topic to find the close connection between miR-149-5p and the TIME.

Sorafenib is approved by the U.S. Food and Drug Administration as a first-line chemotherapy agent for patients with advanced HCC [45]. However, resistance to sorafenib is a tricky issue in treating HCC patients. Niu et al. found that lncRNA NEAT1 sponged miR-149-5p, leading to sorafenib resistance in HCC cells through the NEAT1/miR-149-5p/AKT1 axis [46]. It was suggested that miR-149-5p has the potential to be a target for reversing sorafenib resistance.

MiR-149-5p was also involved in various prognostic models for HCC. Wang et al. constructed a protein-coding gene (PCG)-miRNA signature prognostic model including PCGs and three miRNAs (hsa-miR-149-5p, hsa-miR-424-5p, and hsa-miR-579-5p). The prognostic model had better survival prediction than TNM staging (AUC = 0.72/0.64/0.62) [47]. Fang et al. constructed a p53-related prognostic model containing six miRNAs (has-miR-26a-1-3p, has-miR-188-5p, has-miR-212-5p, has-miR-149-5p, and has-miR-105-5p). Concerning survival prediction ability, especially 5-year survival prediction ability, the model was more advantageous than TNM staging (AUC = 0.76) [48]. Chen et al. used the TCGA database to screen and construct a metastasis-related miRNA (MRM) prognostic model. The model consisted of seven metastasis-associated miRNAs (miR-140-3p, miR-9-5p, miR-942-5p, miR-324-3p, miR-29c-5p, miR-551a, and miR-149-5p), and the low-risk group had longer overall survival (OS) than the high-risk group [49]. Chen et al. showed that three miRNAs (miR-137, miR-149-5p, and miR-561-5p) were associated with lung metastasis in HCC patients [50].

All the prognostic models mentioned above demonstrated better efficiency and specificity, and considering that miR-149-5p is differentially expressed in HCC and normal tissues, it may be a good predictor of HCC patients’ prognosis. MiR-149-5p could also be combined with other HCC predictors, such as AFP, to demonstrate better predictive efficacy, and more studies are needed to explore the possibility.

Overall, miR-149-5p acts as a tumor suppressor in HCC, and targeting its upstream and downstream regulators could be a possible way to inhibit HCC progression. In addition, miR-149-5p remains to be further explored as a predictor of HCC in clinical practice and its potential in the TIME (Figure 1).

#### 3.5.2. Gastric Cancer

GC is the fifth most common malignancy worldwide and the third leading cause of cancer death [51].

MiR-149-5p is mainly regulated by circRNAs and lncRNAs in GC. Zhang et al. reported that miR-149-5p could inhibit GC proliferation, migration, and invasion by binding AKT1 and regulating the AKT1/mTOR axis. CircNRIP1 sponged miR-149-5p and could be transmitted through exosomal communication between GC cells to promote GC metastasis [52]. Hui et al. showed that circNHSL1 was highly expressed in GC tissues, cells, and GC-cell-derived exosomes. CircNHSL1 regulated YWHAZ expression by sponging miR-149-5p, while the knockdown of circNHSL1 inhibited GC migration, invasion, and glutaminolysis [53]. Yang et al. demonstrated that circ_0044516 sponged miR-149-5p and regulated human antigen R (HuR), thus promoting GC-cell proliferation, migration, invasion, and tumor growth [54]. Shao et al. found that circ-DONSON was highly expressed in GC tissues and cells. Circ-DONSON promoted GC-cell proliferation, metastasis, and angiogenesis through the circ-DONSON/miR-149-5p/LDHA axis [55]. Jin et al. found that DNA2-gene-locus-derived circDNA2 was highly expressed in GC tissues and correlated with lymphatic infiltration in patients. Pull-down experiments suggested that circDNA2 promoted GC-cell growth and lymphatic metastasis by sponging miR-149-5p, whose target gene is CCDC6. At the same time, GC patients with lower miR-149-5p expression had shorter survival and weaker tolerance to chemotherapy, suggesting a protective role for miR-149-5p in GC [56]. Cui et al. demonstrated that lncRNA LINC01420 could bind miR-149-5p and promote GC proliferation, migration, and invasion [57]. Li et al. identified a novel lncRNA, SLCO4A1-AS1; it bound miR-149-5p and regulated its target gene, STAT3, in GC, enhancing GC-cell viability, migration, and invasion ability [58]. Qin et al. found that lncRNA OGFRP1 promoted GC-cell proliferation and inhibited apoptosis by regulating the miR-149-5p/MAP3K3 axis [59]. Liu et al. found that circ_0000654 and INHBA were upregulated in GC, and miR-149-5p was downregulated in GC. Circ_0000654 bound miR-149-5p, thus targeting INHBA to promote GC-cell growth [60].

Overall, miR-149-5p also acts as a tumor suppressor in GC, and its upstream circRNAs and lncRNAs can be used as potential targets to inhibit GC progression.

In addition, Qin et al. also found that OGFRP1 overexpression inhibited the radiotherapy sensitivity of GC cells by regulating the downstream miR-149-5p/MAPK3 axis, and that MAPK3 overexpression counteracted the inhibition of GC-cell sensitivity to radiotherapy, caused by OGFRP1 [59]. This study revealed the possible role of miR-149-5p in the radiotherapy sensitization of cancer cells, and more studies could focus on the role of miR-149-5p in radiotherapy sensitization in the future.

Similarly, miR-149-5p could be involved in constructing a prognostic model in stomach adenocarcinoma (STAD). Chen et al. recruited 246 subjects (130 STAD patients and 116 healthy patients) to design a three-phase study and constructed a kit to detect four miRNAs (miR-125b-5p, miR-196a-5p, miR-149-5p, and miR-1-3p) in serum (AUC = 0.892) [61]. Liu et al. identified nine STAD-related datasets from GEO. They screened seven miRNAs that might serve as potential markers for the early diagnosis of STAD patients (miR-455-3p, miR-135b-5p, let-7a-3p, miR-195-5p, miR-204-5p, miR-149-5p, and miR-143-3p) [62]. Kurata et al. identified miR-149-5p as being down-regulated in GC and resistant to adaptation using a microRNA-focused CRISPR-Cas9 library [63].

Meanwhile, Chen et al. showed that miR-149-5p expression was downregulated in STAD serum and could be used as a non-invasive biomarker for STAD. The results of this study also showed the predictive factor potential of miR-149-5p and more studies could focus on its predictive efficiency in different human cancers.

In summary, miR-149-5p played a tumor-suppressive role in GC and was involved in constructing many prognostic models. It also had the potential to become a biomarker for non-invasive diagnosis (Figure 1).

#### 3.5.3. Colorectal Cancer

CRC ranks third in cancer incidence and second in mortality, according to GLOBOCAN 2018 [39].

In CRC, miR-149-5 is regulated by lncRNAs and circRNAs. Lian et al. found that in colorectal cancer, LINC00460 sponged miR-149-5p and regulated CUL4A. Meanwhile, LINC00460/miR-149-5p/CUL4A and LINC00460/EZH2/KLF2 crosstalk could promote CRC tumorigenesis and progression [64]. Wang et al. found that lncRNA PCAT-1 promoted CRC-cell proliferation, migration, and invasion and inhibited CRC-cell apoptosis by targeting miR-149-5p [65]. Ma et al. found that circ5615 could act as a ceRNA for miR-149-5p, upregulating β-actin stabilizing regulator TNKS and activating the Wnt/β-actin pathway to promote CRC progression [66]. Chen et a. found that circTNNA1 sponged miR-149-5p and up-regulated its target gene, FOXM1, promoting CRC-cell proliferation and invasion [67]. Ruan et al. found that LINC00460 acted as a ceRNA for miR-149-5p, regulating the expression of its downstream target gene, BGN, and promoting CRC tumor metastasis [68]. Qu et al. found that lncRNA DLGAP1-AS1 regulated TGFB2 by targeting miR-149-5p to promote CRC-cell proliferation, invasion, and inhibition of apoptosis in vitro and promote tumor growth in vivo [69].

Overall, miR-149-5p also played a tumor-suppressive role in CRC, while its upstream circRNAs and lncRNAs promote CRC progression by suppressing its expression.

Oxaliplatin is the first-line treatment for CRC worldwide, yet intrinsic and acquired forms of resistance to oxaliplatin are a major impediment limiting treatment success [70]. MiR-149-5p was found to be involved in the mechanism of CRC resistance to oxaliplatin. Meng et al. found that LINC00460 expression was upregulated in oxaliplatin-resistant SW480 cells. LINC00460 could upregulate p53 expression by targeting miR-149-5p/miR-150-5p. In addition, mutated p53 genes could positively feedback, and the expression of LINC00460 was upregulated, forming the LINC00460-miR-149-5p/miR-150-5p-MUT p53 axis, leading to drug resistance in CRC [71]. Fluorouracil (5-FU) is often used in combination with oxaliplatin in treating patients with metastatic CRC. Qu et al. found that miR-149-5p is involved in the 5-FU resistance mechanism in CRC and that the lncRNA DLGAP1-AS1/miR-149-5/TGFB2 axis led to 5-FU resistance in CRC cells [69].

These studies indirectly suggested that miR-149-5p played a suppressive role in the mechanism of chemoresistance in CRC, and this role was achieved by being regulated by its upstream lncRNAs. Qi and her colleagues also summarized eight lncRNAs involved in oxaliplatin resistance of CRC and asserted that the depletion of lncRNAs can re-sensitize CRC to oxaliplatin drugs [72]. Thus, lncRNA inhibitors in combination with conventional chemotherapy modalities appear to be feasible. In the future, targeting lncRNAs in CRC treatment needs further study.

Patients with KRAS mutations in CRC usually have a poor prognosis. Assessing miRNA expression levels in tissue samples from CRC patients, Milanesi et al. found that miR-149-5p and seven other miRNAs were specifically down-regulated in CRC tissue samples with KRAS mutations. This study could help discover new molecular biomarkers of clinical relevance in CRC [73].

However, there are no specific studies revealing the specific mechanism of miR-149-5p in CRC with KRAS mutations. Further studies could explore whether miR-149-5p could act as a tumor suppressor by targeting KRAS genes.

In summary, miR-149-5p was reported to act as a tumor-suppressive factor in CRC and to be involved in the resistance to chemotherapy. Further studies could focus on the mechanisms of miR-149-5p in KRAS-mutated CRC (Figure 1).

#### 3.5.4. Oral Cancer

According to GLOBOCAN 2018, the number of lip and oral cancer cases worldwide is approximately 354,864, ranking it as the fourth most common malignancy [39].

In tongue squamous cell carcinoma (TSCC), Chen et al. found that miR-149-5p could inhibit SP1 protein expression and suppress TSCC-cell proliferation, migration, and invasion [74]. Lai et al. found that SP1 mRNA was negatively correlated with miR-149-5p in TSCC tumor tissues and adjacent normal tissues. They also found that carrying the SP1 rs1353058818 locus deletion allele was a high-risk factor for TSCC [75].

Thus, combining these two studies, it is not difficult to find that miR-149-5p plays a tumor-suppressive role in TSCC.

In oral squamous cell carcinoma (OSCC), cisplatin-based chemotherapy is the first-line treatment, but only a portion of OSCC patients benefit from it because of the resistance to cisplatin [76]. Luo et al. found that miR-149-5p promoted cisplatin chemosensitivity by targeting TGF-β and attenuated cell proliferation, apoptosis, migration, and invasion in OSCC [77].

In addition, miR-149-5p was regulated by circRNAs and lncRNAs in OSCC. Qiu et al. found that circBICD2 could regulate IGF2BP1 expression by targeting miR-149-5p, promoting OSCC-cell proliferation, migration, invasion, and glutaminolysis, inhibiting apoptosis [78]. Lv et al. found that lncRNA DLEU1 inhibited OSCC-cell proliferation, migration, and attack and induced OSCC-cell apoptosis by regulating the miR-149-5p/CDK6 axis [79].

In summary, miR-149-5p is a tumor suppressor in both TSCC and OSCC; it also promotes the sensitivity to cisplatin in OSCC (Figure 1).

#### 3.5.5. Esophageal Cancer

Esophageal cancer (EC) is the sixth leading cause of cancer death worldwide [39].

In esophageal squamous cell carcinoma (ESCC), Xu et al. demonstrated in vitro that circ_0000654 indirectly activated the IL-6/STAT3 pathway by binding to miR-149-5p and regulated the proliferation, migration, invasion, and apoptosis of ESCC cells. Circ_000654 also promoted the proliferation and metastasis of ESCC cells in vivo [80]. Li et al. found that lncRNA DRAIC in EC promoted EC-cell proliferation and invasion and inhibited EC-cell apoptosis and autophagy through the miR-149-5p/NFIB axis [81].

In summary, miR-149-5p acts as a tumor suppressor in EC and is regulated by circRNAs (Figure 1).

### 3.6. Cancers of the Respiratory System

#### 3.6.1. Lung Cancer

Lung cancer is the most common cancer worldwide and the leading cause of cancer deaths, with approximately 1.8 million people dying from lung cancer worldwide in 2018 [82]. Lung cancer is histologically divided into non-small cell lung cancer (NSCLC) and small cell lung cancer (SCLC).

The roles in lung cancer are still controversial. In NSCLC, miR-149-5p mainly acts as a tumor-suppressive factor. However, in LUAD, miR-149-5p was reported to act as an oncogenic factor in some cases, while in others, miR-149-5p was also found to act as a tumor-suppressive factor in LUAD.

In NSCLC, miR-149-5p is regulated by upstream lncRNAs and circRNAs. Li et al. found that miR-149-5p expression was down-regulated in NSCLC, and lncRNA PCAT-1 promoted the growth of NSCLC cells by acting as ceRNA for miR-149-5p and upregulating the expression of its downstream target gene, LRIG2 [83]. Liu et al. found that lncRNA HNF1A-AS1 sponged miR-149-5p to regulate CDK6 expression and promote the proliferation, migration, and invasion of NSCLC cells [84]. Chen et al. found that the miR-149-5p/MyD88 axis promoted NSCLC-cell stemness. Ursolic acid can attenuate this effect by inhibiting this axis [85]. Zhou et al. demonstrated in vitro and in vivo that lncRNA MIAT sponged miR-149-5p and up-regulated FOXM1 expression, promoting NSCLC-cell proliferation, invasion, and migration in vitro and tumor growth in vivo [86]. Li et al. found that lncRNA HOTAIR could act as ceRNA for miR-149-5p and regulate HNRNPA1 expression, promoting NSCLC-cell growth, migration, and invasion [87]. Wei et al. found that circ-FOXM1 expression was upregulated in NSCLC and could target miR-149-5p to upregulate ATG5 expression levels, thus promoting cell viability, migration, and autophagy and inhibiting apoptosis in NSCLC [88]. Sun et al. found that B3GNT3 was upregulated in lung cancer and promoted cell proliferation and invasion in vitro. At the same time, miR-149-5p could target B3GNT3 to inhibit its expression and antagonize the tumorigenic effects of B3GNT3 [89].

Overall, miR-149-5p plays a tumor-suppressive role in NSCLC, and its upstream lncRNAs or circRNAs could be targeted to block the progression of NSCLC. As mentioned above, the study by Chen and his colleagues also confirmed the feasibility of this strategy [85].

MiR-149-5p is also involved in the mechanisms of drug resistance in NSCLC. In a previous study, lncRNA HOTAIR enhanced NSCLC resistance to cisplatin by targeting miR-149-5p and regulating DCLK1 expression, suggesting that miR-149-5p promoted NSCLC resistance to cisplatin [90]. In China, Nakano et al. found that LINC00460 acted as ceRNA for miR-149-5p and promoted IL-6 expression, thus inducing epithelial–mesenchymal transition (EMT) and promoting EGFR-TKI resistance in LUAD [91].

Overall, miR-149-5p and its upstream and downstream regulators could be the target to enhance the sensitivity to cisplatin in NSCLC or LUAD. In addition, as Maria Konoshenko et al. summarized, miRNAs could be the predictor of lung cancer resistance and sensitivity to cisplatin; perhaps, further research could explore the predictive roles of miR-149-5p in cisplatin resistance in lung cancer [92].

Tian et al. found that exosome miR-149-5p was an independent risk factor for LUAD and that its upregulation promoted tumor-cell growth and inhibited apoptosis [93]. This suggested that exosome miR-149-5p might be a reliable biomarker for LUAD.

In summary, the role of miR-149-5p in lung cancer is still controversial, and further studies are needed to explore its specific role in lung cancer. More importantly, exosome miR-149-5p may be a novel marker to predict cancer prognosis (Figure 2).

#### 3.6.2. Nasopharyngeal Cancer

In 2018, there were approximately 18.1 million new nasopharyngeal carcinoma (NPC) cases and 9.6 million deaths worldwide [39]. Kong et al. demonstrated that LINC00460 acted as ceRNA for miR-149-5p in NPC and upregulated IL-6, thus promoting the progression of NPC [94] (Figure 2).

### 3.7. Cancers of the Reproductive System

#### 3.7.1. Breast Cancer

According to the 2020 Global Cancer Statistics, breast cancer (BC) has 2.3 million new cases in women every year, surpassing lung cancer, and has become the most common cancer in women. It is also the leading cause of cancer death, posing a severe threat to the life and health of women worldwide [95].

Paclitaxel (PTX) is a first-line drug for treating BC, but prolonged exposure to PXT causes patients to develop inevitable drug resistance, thereby compromising efficacy. Xiang et al. showed that a pentacyclic triterpenoid UA reversed PTX resistance by upregulating miR-149-5p expression levels. In addition, the overexpression of miR-149-5p was shown to effectively enhance PTX-induced apoptosis by inhibiting the expression of MyD88 and the PI3K/Akt signaling pathway, thereby reversing BC drug resistance [96].

The protective effect of miR-149-5p has also been demonstrated in trastuzumab. Trastuzumab is currently the treatment of choice for HER2-overexpressing BC, but its resistance is gradually increasing [97]. Tian and colleagues found that propofol, a common intravenous anesthetic, enhanced the sensitivity of BC cells to HER2 overexpression by epigenetically upregulating IL-6 and miR-149-5p expression in HER2-overexpressing cells, thereby achieving anticancer effects [98]. One study found that miR-149-5p expression was decreased in BC cells and that circSEPT9 promoted glutamine uptake and cell proliferation by sponging miR-149-5p expression to play an oncogenic role in BC malignant progression, providing a novel mechanism for BC development [99]. Qi and co-workers found that circ_0072995 promoted the malignant cellular phenotype and anaerobic glycolysis in BC by sponging miR-149-5p to upregulate SHMT2 [100]. Li et al. found that miR-149-5p expression was reduced in the tissues and cells of triple-negative breast cancer (TNBC). miR-149-5p was found to inhibit cell proliferation, migration, and invasion. circ_0041732 could regulate miR-149-5p by interacting with miR-149-5p FGF5 expression and by inhibiting the apoptosis of TNBC cells [101]. The study by Maimaiti et al. demonstrated that circFAM64A sponged miR-149-5p to increase CDT1 expression, thereby promoting the proliferation, migration, and invasion of TNBC cells and facilitating cell-cycle progression [102].

In short, miR-149-5p can restore the sensitivity to PTX and trastuzumab-resistant BC cells, play a negative role in the progression of BC malignancy through several pathways, and play the role of tumor suppressor (Figure 2).

#### 3.7.2. Cervical Cancer

Cervical cancer has become the fourth most commonly diagnosed cancer in women and the fourth leading cause of cancer-related death in women. It is estimated that there were 570,000 new cases and 310,000 cervical-cancer-related deaths worldwide in 2018 [39].

Based on microRNA sequencing and bioinformatics analyses, investigators identified miR-149-5p as a potential regulator of HPV-positive cervical cancer. Researchers found that the downregulation of miR-149-5p in HPV-positive cervical squamous cell carcinoma (CESC) tissues was 2.3 times lower than that in normal cervical tissues. The identification of DEMs and their targets in the miRNA-TF gene network showed that miR-149-5p was the highest, targeting 15 molecules. These target genes may be involved in many pathological processes of HPV-positive CESC, such as the upregulation of SMC4. They may play an important role in HPV-positive CESC by participating in the E6-mediated p53/Mir-149-5p pathway [103]. Shao et al. found that the expression of hsa_circ_0075341 was abnormally upregulated in cervical cancer patients, which was associated with tumor size, FIGO stage progression, and lymph-node metastasis. Hsa_circ_0075341 enhanced cervical-cancer-cell proliferation and invasion by inhibiting miR-149-5p and upregulating AURKA [104]. Meanwhile, another study confirmed similar findings, i.e., the knockdown of circ_0011385 inhibited cervical-cancer-cell proliferation, migration, invasion, and induced apoptosis. Circ_0011385 was involved in promoting the malignant biological behavior of cervical cancer cells by negatively regulating miR-149-5p to elevate SOX4 expression [105]. Recently, an investigator demonstrated that hsa_circ_0011385 could mediate PRDX6 expression by binding to miR-149-5p. Furthermore, miR-149-5p silencing reversed hsa_circ_0011385 knockdown-mediated angiogenesis and the malignant biological behavior of cervical cancer cells [106]. Overall, the possible role of miR-149-5p as an oncogenic repressor factor and its therapeutic effect in cervical cancer requires further study.

In brief, miR-149-5p, as a key molecule in the pathogenesis of hsa_circ_0075341 and hsa_circ_0011385, plays a tumor suppressor role in cervical cancer. Meanwhile, miR-149-5p is a key molecule in the pathogenesis of CESC and provides a new target for treatment. However, its mechanism in HPV-positive CESC still needs to be further explored (Figure 2).

#### 3.7.3. Ovarian Cancer

Ovarian cancer is one of the most common types of gynecological malignancies, with more than 310,000 new cases and nearly 210,000 new deaths in 2020 [95]; according to statistics, ovarian cancer is one of the most common types of gynecological malignancies. The treatment of ovarian cancer, especially advanced and recurrent ovarian cancer, is the most difficult challenge of clinical work.

PTX is an antineoplastic drug for ovarian cancer, which is less effective due to cellular resistance [107,108]. Therefore, it is important to explore the mechanism of drug resistance, find effective therapeutic targets, and restore the sensitivity of ovarian cancer to PTX. It was found that circ_CELSR1 could positively regulate SIK2 expression by sponging miR-149-5p, and the inhibition of miR-149-5p could effectively restore the sensitivity to PTX in circ_CELSR1-knockdown PTX-resistant ovarian cancer cells [109]. Sun et al. found that miR-149-5p expression was downregulated in chemo-resistant ovarian cancer tissues and cells and that the overexpression of miR-149-5p inhibited ovarian-cancer-cell growth and promoted apoptosis and cisplatin sensitivity [110]. Xu et al. found that miR-149-5p expression was abnormally higher in chemo-resistant ovarian cancer tissues than in chemo-sensitive and normal tissues. Silencing miR-149-5p could enhance the sensitivity of ovarian cancer cells to cisplatin in vitro and in vivo. Conversely, the overexpression of miR-149-5p increased the chemoresistance of ovarian cancer cells. Mechanistic studies showed that miR-149-5p directly inhibited two critical proteins in the Hippo pathway, MST1 and SAV1, thus promoting chemoresistance to cisplatin in ovarian cancer cells [111].

In conclusion, the emerging role of miR-149-5p in ovarian cancer is gradually being discovered; miR-149-5p can effectively reverse PTX resistance, but its role in cisplatin-resistant ovarian cancer is controversial and requires further study.

#### 3.7.4. Endometrial Cancer

Endometrial carcinoma (EC) is a common gynecological tumor. Recently, the five-year survival rate for metastasis-free EC has increased to 74–91% due to the development of treatment and adjuvant therapies. However, for patients with recurrent EC, available treatments are minimal. Therefore, there is an urgent need to identify the optimal treatment for patients with EC. Liu’s study suggested that hsa_circ_0061140 may act as a tumor promoter in EC by working as a molecular sponge for miR-149-5p and inducing STAT3 expression, providing new ideas for the treatment of EC [112].

There are few studies on miR-149-5p in EC. Whether miR-149-5p affects the occurrence and development of EC through some pathways needs more research (Figure 2).

#### 3.7.5. Prostate Cancer

Prostatic cancer (PCa) is the second most common cancer among men and the fifth leading cause of cancer-related deaths in men worldwide. It is estimated that there were nearly 1.4 million new cases and 375,000 deaths due to PC worldwide in 2020 [39,95].

In 2019, it was reported that Fuzheng Yiliu decoction, a traditional Chinese medicine preparation, could inhibit tumor growth and change the expression of miRNAs in PCa tissues, including miR-149-5p. These miRNAs enhanced the therapeutic effect by modifying the PI3K-Akt pathway in cancers [113]. Experiments by Ma et al. demonstrated that miR-149-5p expression was reduced in PCa tissues and cells and inhibited PCa-cell viability, proliferation, and migration ability by suppressing RGS17 expression [114]. In addition, Temiz. et al. found that the overexpression of miR-149-5p in PCa cell lines downregulated CCT3 expression, which led to the disruption of intracellular ROS homeostasis, altered distribution of free amino acids in energy metabolism, and promoted apoptosis in tumor cells [115].

In summary, miR-149-5p plays a positive role in enhancing therapeutic efficacy, inhibiting tumor-cell growth and promoting apoptosis and is a critical tumor suppressor in PCa. 

### 3.8. Cancers of the Urological System

#### 3.8.1. Kidney Cancer

Renal cell carcinoma (RCC) is the most common renal tumor, and approximately 70–80% of RCC is classified as a clear-cell type [116]. RCC is diagnosed in over 270,000 patients yearly and accounts for approximately 3% of adult malignancies. Incidence of and mortality from RCC increase by 2–3% per decade [117].

Jin et al. first elucidated the expression and function of miR-149-5p in RCC. The expression of miR-149-5p was significantly down-regulated in RCC tissues compared with normal kidney tissues. The restoration of miR-149-5p expression with synthetic mimics inhibited RCC-cell proliferation and migration and promoted apoptosis [118]. Clear-cell renal cell carcinoma (ccRCC) is the most common subtype of RCC with a high incidence and poor prognosis [119]. Okato’s team investigated the role of the double chain of pre-miR-149 in ccRCC. They showed that miR-149-5p and miR-149-3p double chains exhibited anti-tumor effects by acting on FOXM1 to inhibit the proliferation, migration, and invasion of renal cancer cells [120]. In addition, Xie et al. constructed a signature containing four miRNAs (miR-21-5p, miR-9-5p, miR-149-5p, and miR-30b-5p) associated with the survival of ccRCC patients, which could be used as a prognostic biomarker for ccRCC [121]. These results suggest that miR-149-5p may be a tumor suppressor in RCC and may act as a prognostic marker for RCC.

#### 3.8.2. Bladder Cancer

Bladder cancer is the fifth most common cancer worldwide and usually occurs in the bladder epithelium [122]. It is estimated that there are approximately 150,000 new cases of bladder cancer and over 50,000 deaths each year [123].

MiR-149-5p is a suppressor of various cancers and plays a tumor suppressor role in bladder cancer. Wang et al. found that circRNA_100146 promoted RNF2 expression by sponging miR-149-5p, promoting bladder-cancer-cell proliferation, migration and invasion. In addition, Lin and other researchers found that the urinary expression levels of miR-149-5p were significantly higher in bladder cancer patients than in healthy controls. An analysis of The Cancer Genomics Atlas (TCGA) database showed that a high expression of miR-149-5p (crude hazard ratio (CHR), 1.52; 95% confidence interval (CI) 1.00–2.33; P = 0.05) was significantly associated with poor overall survival rate in bladder cancer patients. A multivariate logistic analysis showed that a high expression of miR-149-5p (adjusted HR (AHR), 1.70; 95% CI, 1.11 to 2.61; P = 0.015) was still significantly associated with poor overall survival. Therefore, miR-149-5p in urine may be a potential biomarker for the non-invasive prognosis of bladder cancer patients [124] (Figure 2).

## 4. Discussion

In summary, the roles of miR-149-5p in cancers are two-sided; it acts as a tumor suppressor in TC, BC, NSCLC, and urinary- and digestive-system cancers, while it acts as an oncogenic factor in LUAD, ovarian cancer, and hematological malignancies (Table 1). However, in most cases, miR-149-5p is a tumor-suppressive factor in cancers. MiR-149-5p plays an important role in the tumorigenesis, development, and prognosis of human cancers through the regulation of its upstream circRNAs and lncRNAs. Due to the small sample size and to existing studies being focused on targeting exploration and basic mechanism verification, the specific circRNA–miRNA–mRNA axis is still unclear. How to overcome the existing difficulties and successfully construct a miR-149-5p functional network centered on circRNA–miRNA–mRNA axis in vivo is the core of current miR-149-5P research.

In addition, research of miRNA-based biomarkers in cancer has made rapid progress. Many studies showed that miRNAs are easy to detect and are highly sensitive as biomarkers [8,125]. A large number of research studies showed that the expression of miR-149-5p in the body fluids of tumor patients is different from that of healthy people [31,62,124]. Using this feature and the fusion of bioinformatics, genomics, and other disciplines, as well as biological engineering, miR-149-5p could be assumed to be a non-invasive potential biomarker in the early diagnosis of tumors and could even be developed as an early tumor detection biosensor, improving the early cancer detection rate and perfecting the malignant tumor stage and the tumor prognosis evaluation index [126]. In addition, the performance of miR-149-5p in tumor therapy and drug resistance reflects its great potential as a target for tumor-drug-targeted therapy [109].

In conclusion, miR-149-5p plays an important role in human cancers and is a potential biomarker for tumor diagnosis, treatment, and prognosis. It is necessary to combine basic research with clinical practice, deeply study the corresponding mechanism, strive to apply the experimental results to clinical practice, and guide clinical treatment work.

## Figures and Tables

**Figure 1 ijms-23-10823-f001:**
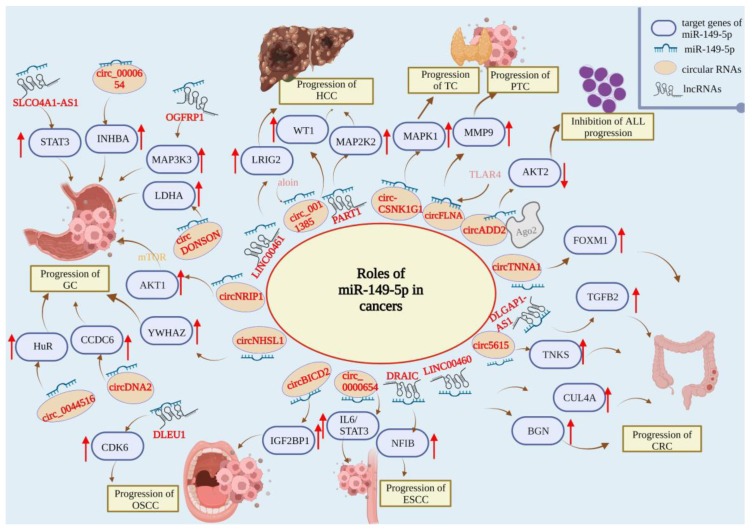
Roles of miR-149-5p in papillary thyroid cancer (PTC), thyroid cancer (TC), acute lymphoblastic leukemia (ALL), hepatocellular carcinoma (HCC), gastric cancer (GC), colorectal cancer (CRC), oral squamous cell carcinoma (OSCC), and esophageal squamous cell carcinoma (ESCC).

**Figure 2 ijms-23-10823-f002:**
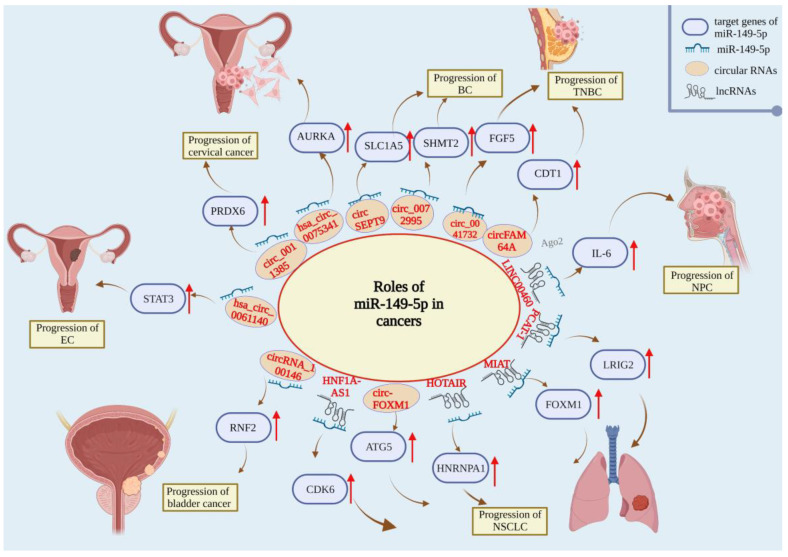
Roles of miR-149-5p in non-small cell lung cancer (NSCLC), nasopharyngeal carcinoma (NPC), breast cancer (BC), triple-negative breast cancer (TNBC), cervical cancer, endometrial carcinoma (EC), and bladder cancer.

**Table 1 ijms-23-10823-t001:** Roles of miR-149-5p in different human cancers.

Human System	Tumor Type	Upstream Regulator	Target Gene	Mechanism	Role	References
Endocrine system	TC	rs2292832	NA	Inhibits the local progression of PTC	Tumor suppressor	[26]
		ciecFLNA	MMP9	Promotes the invasion and migration of PTC cells		[27]
		NA	CCL18	Attenuates the lymph-node metastasis of TC cells		[28]
		circ-CSNK1G1	MAPK1	Inhibits cell proliferation and invasion		[29]
		NA	GIT1	Inhibits the proliferation and invasion of MTC cells		[30]
Hematological system	Leukemia	NA	FASLG	Activates FADD and caspase and promotes apoptosis	Tumor suppressor	[31]
		hsa_circ_0120872	AKT2	Inhibits cell proliferation and induces cell apoptosis		[32,33]
Skeletal system	Chondrosarcoma	NGF	PI3K/Akt/mTOR	Inhibits the occurrence and metastasis of chondrosarcoma	Tumor suppressor	[38]
Digestive system	HCC	SNHG8	NA	Inhibits HCC proliferation, invasion, and lung metastasis	Tumor suppressor	[40]
		LINC00461	LRIG2	Inhibits the proliferation, migration, and invasion of HCC cells		[41]
		NEAT1	AKT1	Reverses the resistance of HCC cells to sorafenib		[46]
		PART1	MAP2K1	Inhibits the proliferation, migration, and invasion of HCC cells		[42]
		circ_0011385	WT1	Inhibits cell proliferation, invasion, and tumor growth and promotes the apoptosis and autophagy of HCC		[43]
		M2Macrophages	MMP9	Inhibits HCC progression		[44]
	GC	circNRIP1	AKT1	Inhibits GC proliferation, migration, and invasion	Tumor suppressor	[52]
		circNHSL1	YWHAZ	Inhibits GC migration, invasion, and glutaminolysis		[53]
		circ_0044516	HuR	Inhibits GC-cell proliferation, migration, and invasion		[54]
		circ-DONSON	LDHA	Inhibits GC-cell proliferation, metastasis, and angiogenesis		[55]
		circDNA2	CCDC6	Inhibits GC growth and lymphatic metastasis		[56]
		LINC01420	NA	Inhibits GC-cell proliferation, migration, and invasion		[57]
		SLCO4A1-AS1	STAT3	Inhibits GC-cell migration and invasion		[58]
		OGFRP1	MAP3K3	Inhibits GC-cell proliferation and enhances apoptosis and radiosensitivity		[59]
		circ_0000654	INHBA	Inhibits the growth of GC cells		[60]
	CRC	LINC00460	CUL4A	Inhibits CRC tumorigenesis and progression	Tumor suppressor	[64]
		PCAT-1	NA	Inhibits the proliferation, migration, and invasion of CRC cells and promotes the apoptosis of CRC cells		[65]
		circ5615	TNKS	Inhibits CRC tumorigenesis and progression		[66]
		circTNNA1	FOXM1	Inhibits the proliferation and invasion of CRC cells		[67]
		LINC00460	BGN	Inhibits CRC tumor metastasis		[68]
		DLGAP1-AS1	TGFB2	Inhibits the proliferation and invasion of CRC cells and inhibits apoptosis		[69]
		LINC00460	p53	Leads to oxaliplatin resistance		[71]
	Oral cancer	NA	SP1	Inhibits TSCC-cell proliferation, migration, and invasion	Tumor suppressor	[74]
		NA	TGF-β	Promotes the chemical sensitivity of cisplatin and reduces cell proliferation, apoptosis, migration, and invasion		[77]
		circBICD2	IGF2BP1	Promotes OSCC-cell proliferation, migration and invasion; glutamine dissolution; and apoptosis		[78]
		DLEU1	CDK6	Inhibits OSCC-cell proliferation, migration, and invasion and induces OSCC-cell apoptosis and G1 stagnation		[79]
	Esophageal cancer	circ_0000654	IL-6/STAT3	Inhibits the proliferation, migration, invasion, and apoptosis of ESCC cells	Tumor suppressor	[80]
		DRAIC	NFIB	Inhibits EC-cell proliferation and invasion and promotes esophageal-cancer-cell apoptosis and autophagy		[81]
Respiratory system	Lung cancer	PCAT-1	LRIG2	Inhibits the growth of NSCLC cells	Controversial	[83]
		HNF1A-AS1	CDK6	Inhibits the proliferation, migration, and invasion of NSCLC cells		[84]
		Ursolic acid	MyD88	Promotes NSCLC-cell stemness		[85]
		MIAT	FOXM1	Inhibits proliferation, invasion, and migration of NSCLC cells		[86]
		HOTAIR	HNRNPA1	Inhibits the growth, migration, and invasion of NSCLC cells		[87]
		circ-FOXM1	ATG5	Inhibits cell migration and autophagy in NSCLC		[88]
		HOTAIR	DCLK1	Promotes cisplatin resistance in NSCLC		[90]
		LINC00460	IL-6	Induces EMT and promotes EGFR-TKI resistance		[91]
		NA	B3GNT3	Inhibits the proliferation and invasion of lung cancer cells		[89]
	NPC	LINC00460	IL-6	Inhibits the malignant progression of NPC	Tumor suppressor	[94]
Reproductive system	BC	NA	My88	Reverses PXT resistance	Tumor suppressor	[96]
		NA	IL-6	Reverses trastuzumab resistance		[98]
		circSEPT9	SLC1A5	Inhibits glutamine uptake and inhibits cell proliferation		[99]
		circ_0072995	SHMT2	Inhibits malignant phenotype and anaerobic glycolysis in breast cancer cells		[100]
		circ_0041732	FGF5	Inhibits cell proliferation, migration, invasion, and vascular formation and increased cell apoptosis		[101]
		circFAM64A	CDT1	Inhibits TNBC proliferation, cell-cycle migration, and invasion		[102]
	Cervical cancer	has_circ_0075341	AURKA	Reduces the proliferation and invasion of cervical cancer cells	Tumor suppressor	[104]
		circ_0011385	SOX4	Inhibits the proliferation, migration, and invasion of cervical cancer cells and inhibits apoptosis		[105]
		hsa_circ_0011385	PRDX6	Reverses the angiogenesis and malignant biological behavior of cervical cancer cells		[106]
	Ovarian cancer	circ_CELSR1	SIK2	Inhibits paclitaxel resistance and inhibits malignant cell progression	Controversial	[109]
		NA	MST1 and SAV1	Promotes chemotherapy resistance of ovarian cancer cells to cisplatin		[111]
		NA	XIAP	Inhibits ovarian-cancer-cell growth and promotes apoptosis and cisplatin sensitivity		[110]
	EC	hsa_circ_0061140	STAT3	Inhibits EC malignant progression	Tumor suppressor	[112]
	PCa	NA	PI3K-Akt	Enhances the therapeutic effect of Fuzheng Yiliu decoction	Tumor suppressor	[113]
		NA	RGS17	Inhibits the active proliferation and migration of PCa cells		[114]
		NA	CCT3	Destroys ROS homeostasis and promotes tumor-cell apoptosis		[115]
Urinary system	RCC	NA	FOXM1	Inhibits the proliferation, migration, and invasion of renal carcinoma cells	Tumor suppressor	[120]
	Bladder cancer	circ_100146	RNF2	Inhibits the proliferation, migration, and invasion of cysts	Tumor suppressor	[124]

TC, thyroid cancer; PTC, papillary thyroid cancer; MTC, medullary thyroid cancer; HCC, hepatocellular carcinoma; GC, gastric cancer; CRC, colorectal cancer; TSCC, tongue squamous cell carcinoma; OSCC, oral squamous cell carcinoma; ESCC, esophageal squamous cell carcinoma; NSCLC, non-small cell lung cancer; EMT, epithelial–mesenchymal transition; NPC, nasopharyngeal carcinoma; BC, breast cancer; PXT, Paclitaxel; TNBC, triple-negative breast cancer; EC, endometrial carcinoma; PCa, prostatic cancer; RCC, renal cell carcinoma; NA, not available.

## Data Availability

Not applicable.

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
