# Peer review of "Tumor-Suppressive and Oncogenic Roles of microRNA-149-5p in Human Cancers"

_ijms, 2022, doi:10.3390/ijms231810823_

Round 1

Reviewer 1 Report

Overview:

  • The current manuscript provides a holistic view of the role of miR149-5p in various cancers, including its functions, mutations, expressions and effect on pathology. This provides great foundation of this particular miRNA in the context of cancers. This review provides substantial amount of experimental evidence from prior investigations. However, the review paper can improve by adding more insights, and improving the ‘physiology section’ to include human relevant examples.

Formatting:

  • Figure 1 - Please adjust the image such that the text fits or wraps within the box.
  • Further add a legend to differentiate the blue boxes, yellow boxes, and miRNA.

Technical Remarks:

  • The manuscript is well drafted, however, the second section needs to be revised for relevance. A number of examples provided were of non-human origin, to describe the physiological roles of miR149-5p. Either provide insight into how these findings might translate into human pathophysiology, or revise these examples in a human context. These are seen in lines 77-79, 97-101.
  • The authors have also provided a substantial amount of evidences from prior investigation, but in certain areas, it is suggested to provide the implications of these findings (what these evidences mean for pathology, what they suggest in general). In Lines 133-135, the authors have added this implication appropriately. However, this is required in lines 95, 147, 174, 204, and 286.
  • To improve the discussion, the authors may include the future visions the authors have for future investigations and the development of miR149-5p as a bio marker for various cancers.
  • Kindly, also include the current knowledge gaps regarding miR149-5p, to encourage investigators to explore certain avenues, through the guidance of your paper.

Reviewer 2 Report

This review restates a lot of information of microRNA-149-5p from other publications. However, these information was note well organized and summarized. I think the authors need to reorganize the whole manuscript, and put more summarized information as well as their own opinions into it.

Also, the figures definitely can be improved. For example, in figure 1," Progression of HCC" is out of the box. 
